

# Implementation of a Multi-resolution Analysis Method to Characterize Multi-Scale Wave Structures in Lidar Data

Samuel Trémoulu[1], Fabrice Chane Ming[1], Sitraka Fabrice Raharimanjato[1], Alain Hauchecorne[2], Sergey Khaykin[2], and Philippe Keckhut[2]

[1]LACy, CNRS/Météo-France, UMR 8105, Université de La Réunion, 97744 Saint-Denis de La Réunion, France
[2]LATMOS-IPSL, CNRS/INSU, UMR 8190, Université de Paris-Saclay, 78280 Guyancourt, France

**Correspondence:** Samuel Trémoulu (samuel.tremoulu@univ-reunion.fr)

**Abstract.** This study introduces a processing method based on multi-resolution analysis (MRA) to characterize the multi-scale structures of gravity waves (GWs) with vertical wavelengths less than 13 km in lidar vertical profiles of temperature and wind in the middle atmosphere. The MRA approach is evaluated against conventional techniques, including polynomial fitting, spectral filtering, and nighttime temporal averaging, and applied to a case study of GWs observed on November 20, 2023. Among these

methods, MRA demonstrates superior performance by enhancing the signal-to-noise ratio through signal decomposition and selective filtering. This targeted filtering improves the detection and extraction of GW-induced perturbations, particularly for dominant vertical wavelengths around 5 km. In terms of GW potential energy (GWPE), the MRA-based method yields values comparable to those derived from the variance method, except at the stratopause, where it estimates nearly twice the GWPE. However, the variance-based estimate remains within the MRA-derived confidence interval, indicating good agreement. In

contrast, the Butterworth low-pass filter produces energy densities an order of magnitude higher than the variance method, suggesting possible overestimation of perturbation amplitudes. Polynomial fitting and nighttime mean methods appear insensitive to small-scale GW structures near the stratopause, where wave dissipation may occur. Beyond energy estimation, the MRA method offers a distinct advantage for analyzing GW propagation and scale interactions due to its multi-scale decomposition capability. It reveals GW features and structures that remain obscured by common techniques, establishing it as a valuable tool

for advancing the study of GW dynamics in the middle atmosphere.

## 1 Introduction

Atmospheric gravity waves (GWs) have become a major focus of research in recent years because of their significant effects on atmospheric dynamics and chemistry, as well as the limitations in resolving small-scale structures in current numerical weather prediction and climate models (Fritts and Alexander, 2003). As they propagate, GWs strongly influence large-scale

atmospheric processes by modulating both horizontal and vertical momentum fluxes (e.g., the Brewer-Dobson circulation, the pole-to-pole circulation). They also play a crucial role in vertically coupling the lower and middle atmosphere.

Over the past four decades, lidars have proven to be invaluable tools for observing and characterizing vertically propagating GWs in the middle atmosphere (e.g., Chanin and Hauchecorne, 1981; Gardner et al., 1993; Wilson et al., 1991; Whiteway





and Carswell, 1995; Chane-Ming et al., 2000; Duck et al., 2001; Rauthe et al., 2008; Yamashita et al., 2009; Alexander

et al., 2011; Kaifler et al., 2015). These studies highlight the capability of lidar observations to infer long-term trends in GW activity. In particular, lidars provide unique, high-resolution dynamical measurements of temperature and wind across the middle atmosphere, with excellent temporal and vertical resolution. Such measurements are widely used to investigate GW propagation, especially in the mesosphere and lower thermosphere (MLT) region (Liu et al., 2009; Placke et al., 2013; Lu et al., 2015, 2017; Kaifler et al., 2015; Chen et al., 2016). However, lidars typically produce one-dimensional, nighttime

vertical profiles, which limits their ability to resolve the horizontal structure and intrinsic properties of atmospheric waves.

Retrieving GW activity from observations requires isolating wave-induced perturbations from the estimated background state. The wave signal itself spans a broad spectral range, including contributions from tides, planetary waves, and GWs. Depending on the observational technique and data processing approach, the fluctuating dynamical component may be attributed primarily to GWs. However, a major challenge lies in effectively distinguishing large-scale GWs from other overlapping wave

types. Over the past several decades, various methods have been developed to extract GW perturbations from lidar-derived temperature and wind profiles by removing the background signal. A commonly used technique involves subtracting a nightly mean profile, considered representative of the background, from each individual profile (Gardner et al., 1989; Rauthe et al., 2008; Ehard et al., 2014). Another widely applied method fits a polynomial function to the measured profiles to isolate the perturbation field (Whiteway and Carswell, 1995; Duck et al., 2001; Hertzog et al., 2001; Alexander et al., 2011). A further

approach to separating GW contributions from large-scale atmospheric motions involves the use of high- or low-pass filters, with the cutoff wavelength determined by the specific characteristics of the filter (Chane-Ming et al., 2000). For GW analysis, high-pass filters are typically applied in the time domain to remove low-frequency components associated with large-scale waves, and in the height domain to eliminate tidal influences (Hirota, 1984; Hirota and Niki, 1985; Eckermann et al., 1995; Hertzog et al., 2001).

The sensitivity of GW detection methods varies across the wave spectrum and is also strongly influenced by the capabilities of the observing instrument, particularly lidar systems, where performance depends on factors such as laser power and collection area. In the literature, GW potential energy density is commonly estimated over a wide spectral range; however, comparing results across studies remains challenging. This difficulty arises from the inability to clearly distinguish variations caused by differences in methodology from those driven by geophysical variability (Ehard et al., 2015). Since the 1980s,

wavelet theory has found widespread application in signal and image processing. More recently, in combination with neural networks, it has gained increasing relevance in the field of machine learning (Guo et al., 2022). Wavelet analysis is particularly well-suited for examining nonstationary, multiscale, wave-like structures, as it enables the resolution of spectral characteristics in both time and space. As such, it is a powerful tool for capturing the dynamics of GWs whose signatures are embedded in lidar-derived temperature and wind perturbations (Chane Ming et al., 2023). Orthogonal discrete wavelets further allow for the

continuous tracking of spectral energy evolution with altitude, making use of the principle of energy conservation to assess vertical variations in GW activity. The linear nature of wavelet analysis also makes it well-aligned with the linear theory of GWs, which describes the wave field as a superposition of monochromatic components. Moreover, wavelet-based methods facilitate the investigation of turbulence-related processes, including energy cascades and nonlinear interactions among struc-



tures at different scales. In this context, the present study introduces a method based on the multiresolution analysis (MRA).
MRA, formalized by Mallat (1989), enables the construction of orthogonal discrete wavelets and is a highly versatile tool that offers several key advantages for the analysis of GWs. It enables efficient signal denoising, multi-scale filtering, and accurate signal reconstruction, making it particularly well-suited for studying the complex, multiscale nature of GWs. GW activity in the middle atmosphere is typically quantified by calculating potential and kinetic energy densities from temperature and wind perturbations, respectively (Li et al., 2023; Brhian et al., 2024; Wüst et al., 2024). These energy densities diagnostics provide
complementary insights into different aspects of GW dynamics, with kinetic energy density generally being more sensitive to low-frequency waves than potential energy (Geller and Gong, 2010).

Beyond perturbation-based methods, Mzé et al. (2014) proposed an alternative approach for estimating potential energy directly from raw lidar photon count profiles. This method builds on the variance-based technique originally developed by Hauchecorne et al. (1994), allowing GW energy to be inferred without requiring explicit background subtraction.

In this paper, we present a method based on MRA to characterize multi-scale GWs in observational data, with a particular focus on lidar measurements. The paper is organized as follows: Section 2 provides a description lidar data and four methods for the analysis of GWs, with an emphasis on MRA; in section 3, all methods are applied to temperature lidar data, and the characteristics as well as the comparative performance of the four methods are discussed, additionally, further applications of the MRA are demonstrated through its implementation on wind lidar data; finally, conclusions and potential applications of the
MRA are outlined.

## 2  Materials and method

### 2.1  Lidar data

Since 2013, the Rayleigh-Mie-Raman (RMR) lidar and the Rayleigh-Mie-Doppler (RMD) lidar have both been operating at the Maïdo Observatory on La Réunion (21°S, 55°E). The RMR lidar provides vertical profiles of temperature spanning the
middle atmosphere, covering altitudes between 30 km and 90 km (Baray et al., 2013). Lidar temperature measurements are useful for the study of the middle atmosphere especially for the characterization of multi-scale dynamical processes such as mesospheric inversion layers and propagating waves (Bègue et al., 2017; Chane Ming et al., 2023). Additionally, horizontal wind velocities are produced by the RMD lidar at heights ranging from 5 km to 60 km (Khaykin et al., 2018). Lidar data are mainly used for long-term monitoring of the middle atmosphere and for calibration and validation (Cal/Val) of campaigns such
the recent Cal/Val of the European Space Agency (ESA) ADM-Aeolus satellite mission for global wind observations (Ratynski et al., 2023).

Rayleigh lidar operates by measuring atmospheric density, which is directly proportional to molecular Rayleigh scattering, and calculates temperature through the downward integration of the hydrostatic law (Hauchecorne and Chanin, 1980). The light source of this lidar consists of two Quanta Ray Nd:Yag lasers. The final wavelength emitted is 355 nm with a pulse repetition at
30 Hz and each pulse delivers 375 mJ. The backscattered signal is collected by a 1.2 m diameter telescope (Gantois et al., 2024). The temperature profile is initialized at the top using a seed temperature from the NRLMSISE-00 empirical atmospheric model





(Picone et al., 2002). Initially, raw Rayleigh-Mie-Raman (RMR) temperature profiles are obtained with a 1-minute integration time and a vertical resolution of 150 meters. To enhance the signal-to-noise ratio, vertical smoothing and time binning are applied according to the scientific objectives. For example, for nightly mean profiles available on the NDACC database, a 2 km vertical smoothing is applied using a Hanning filter to reduce noise and improve data accuracy.

The horizontal wind components are determined by measuring the Doppler shift between the emitted and backscattered light, induced by the projection of molecular or particle velocities along the laser's line of sight, which is inclined off-zenith. To capture both wind components, the laser beam is alternately directed along the zonal and meridional directions with a 45° elevation. A vertical pointing configuration is also employed to establish the zero Doppler shift reference, based on the assumption that vertical wind velocities are negligible (Chanin et al., 1989; Souprayen et al., 1999). The Doppler lidar uses a Nd:YAG laser operating at 532 nm in monomode. The pulse repetition rate of the laser is 30 Hz with 24 W mean energy. The 0.3 m2 telescope of Maido wind lidar is composed of a single rotating mirror, which serves for both the emission and reception. The initial Rayleigh-Mie-Doppler wind profiles are retrieved with a temporal resolution of 5 minutes and a vertical resolution of 200 meters (Khaykin et al., 2016, 2018) The present study uses individual lidar temperature profiles of 15 min integration time and 150 m vertical resolution, along with lidar wind profiles of 5 min integration and 200 m vertical resolution to characterize GWs with vertical wavelengths and observed periods > 1 km and 1 hour respectively. Lidar measurements are done during the 20 November 2023 night between 1543 UTC and 2028 UTC for temperature and between 1528 UTC and 2350 UTC for wind.

## 2.2 Gravity waves analysis techniques

### 2.2.1 The time averaged background profiles method

The nightly mean temperature profile is a simple, robust and widely used method for determining background temperature profiles (Gardner et al., 1989; Rauthe et al., 2008; Ehard et al., 2014). This approach assumes that the timescale of the background influencing temperature profiles is longer than the measurement period, while the timescale of GWs is relatively shorter ( 3-12 hours). However, spectral bands of GWs are difficult to define. Alternatively, the background temperature profile can be determined using a running mean over a time window, typically around 3 hours (Yamashita et al., 2009). In this method, temperature variations with timescales longer than the window are attributed to the background profile and are excluded from the extracted GW spectrum. In our study, the nightly mean background temperature profile is derived by applying a Hanning window, with vertical smoothing performed using a 7.5 km vertical wavelength filter.

### 2.2.2 The polynomial fit method

The background temperature profile can also be separated by fitting an nth-degree polynomial to the temperature profile, representing the background field (Allen and Vincent, 1995). The polynomial order is determined based on the height range of the analysis. If L is the total height range, a 2nd-order polynomial fit removes perturbations with wavelengths greater than 2L. Similarly, 3rd and 4th-order polynomial fits can remove perturbations with wavelengths greater than $L$ and $\frac{2L}{3}$,





respectively. Polynomial fits of order greater than four may remove substantial GW signals from the analysis (Guest et al., 2000). To calculate the GW spectrum for use in numerical models based on radiosonde observations in the lower atmosphere of the Southern Hemisphere, Pfenninger et al. (1999) used different-order polynomials for different atmospheric layers. They applied lower-order polynomials (2nd or 3rd) for the troposphere, where the mean temperature profiles are mostly linear, and higher-order polynomials (above 3) for the stratosphere, where the mean temperature profiles are much more complex.

In our study, the background temperature profile is determined using a 4th-order polynomial fit (with $\lambda_v < \frac{2L}{3}$. In this case, perturbations with vertical wavelengths greater than 26 km are removed from lidar perturbations profiles.

### 2.2.3 The spectral filtering method

Spectral filtering is another common method to study GWs. Contributions of GWs can be separated from the large scale waves by applying a high/low-pass filter of any cutoff wavelength. Generally, in order to remove low frequency components (or large scale waves) such as tides, high-pass filters are used in the time domain (or height domain) (Hirota, 1984; Hirota and Niki, 1985; Eckermann et al., 1995).

To isolate GW-induced perturbations specifically, the filtering function must be carefully selected to ensure an appropriate spectral response. Chane-Ming et al. (2000) applied a high-pass butterworth filter, with a cutoff of 12 km vertical wavelength, on individual temperature profiles to extract GW perturbations. They, hence, limit the vertical wavelength space to a GW field with vertical wavelengths < 12 km. The Butterworth filter is well adapted for the study of GWs due to its flat frequency response in the passband and is minimizing distortions while effectively isolating the desired wave components.

In this study, we apply a 5th-order Butterworth high-pass filter (Ehard et al., 2015) with a cutoff wavelength of 8 km, allowing us to extract GWs with the observed dominant vertical wavelength of 5 km during the studied night.

### 2.2.4 The variance method

The variance method, described by Mzé et al. (2014), is a technique used for calculating directly the potential energy density of GWs by using raw lidar signal, and is robust against data processing errors (Hauchecorne et al., 1994; Khaykin et al., 2015). The signal originates from an incoherent backscatter lidar operating in photon-counting mode. The raw signal is aggregated over small time and vertical intervals, allowing the vertical profile to be decomposed into a smooth mean profile (background signal) and short-scale perturbations. The relative perturbations, defined as $S'(z_i, t_j) = \frac{dS}{\bar{S}}$, may arise from either instrumental noise or atmospheric fluctuations (where $z_i$ and $t_j$ are a given altitude and time, respectively) . To compute the observed variance and instrumental variance, larger time and verticalsinterval $(\Delta T, \Delta Z)$ are considered by grouping multiple elementary intervals such as $\Delta T = \Delta t N_t$ and $\Delta Z = \Delta z N_z$. The observed variance represents the sum of both instrumental and atmospheric variances and is defined as :

$$V_{obs} = \frac{1}{N_t N_z} \sum_{N_z} \sum_{N_t} S'(z_i, t_j)^2 \tag{1}$$





Since the lidar signal in photon-counting mode follows Poisson's distribution, the instrumental variance can be directly de-
rived from the estimated mean signal S, except in cases of signal saturation due to exceptionally strong returns. The atmospheric
variance is then obtained as the difference between the observed and instrumental variances :

$$V_{atm} = V_{obs} - V_{inst} \tag{2}$$

This atmospheric variance serves as an estimator of GW activity in the middle atmosphere and is used in the equation of
GW potential energy density (Wilson et al. 1991) as followed:

$$E_p = \frac{1}{2}\left(\frac{g}{N}\right)^2 V_{atm} \tag{3}$$

Where $V_{atm}$ is the equivalent of $< \rho'/\rho_0 >^2$, hence $< T'/To >^2$ in adiabatic and linear conditions.

### 2.2.5 The multiresolution analysis method

This study focuses on the MRA based method of which the description of this new method is detailed as opposed to the
four previous methods. The multi-resolution analysis (MRA) framework provides the theoretical foundation for the discrete
wavelet transform (DWT), enabling the hierarchical decomposition of a signal into different scales of approximation and detail.
It decomposes a signal into orthonormal bases, capturing both coarse approximations and successive details across multiple
resolution levels. By adding or removing details, MRA enables a smooth transition between high and low resolutions, with
each detail level encoding the differences in information between successive scales. The effectiveness of this approach is further
enhanced by its flexibility in constructing orthogonal or biorthogonal bases tailored to the analyzed signal. When combined
with efficient algorithms from subband coding theory and filter banks, MRA becomes a robust and adaptive tool for signal
analysis (Mallat, 1989; Hubbard, 1998). The original signal is decomposed into successive octave bands (in the dyadic MRA,
most widely used), with each level represented by an approximation of order n and discrete details up to that order, forming a
pyramid-like wavelet decomposition tree, as described by Mallat's algorithm. Hence, the original signal can be decomposed as
followed :

$$s(n) = d_1(n) + d_2(n) + d_3(n) + \cdots + d_i(n) + a_i(n) \tag{4}$$

The vertical resolution is crucial in this process, as adjusting it modifies the octave band limits, thereby altering the vertical
wavelength bands. This adaptability allows for the selective extraction and analysis of GWs at different scales, depending on the
chosen vertical resolution. Our method uses the orthogonal Daubechies wavelet which is mostly used in signal reconstruction.
This wavelet and its corresponding scaling function of order 8 are sufficiently smooth to provide an optimal balance between
computational efficiency and the quality of the decomposed GW signals (Chane-Ming et al., 2000; Chane Ming et al., 2023). All
lidar profiles are oversampled to a vertical resolution of 100 m, however the study focuses on GWs with vertical wavelengths
of 0.8-12.8 km. The 100 m vertical sampling provides different spectral domains corresponding to the wavelength bands :
0.2-0.4 km, 0.4-0.8 km, 0.8-1.6 km, 1.6-3.2 km, 3.2-6.4 km, 6.4-12.8 km and >12.8 km as shown in Fig. 1.





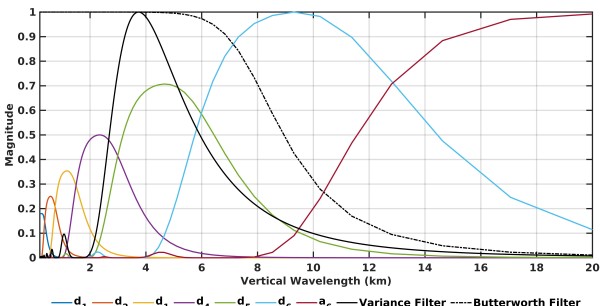

**Figure 1.** Filter bank for 8th order Daubechies wavelet (dark blue : $d_1$, orange : $d_2$, yellow : $d_3$, purple : $d_4$, green : $d_5$, lightblue : $d_6$, brown : $a_6$), Filter response for the variance method for $N_z = 21$ which represent the number of elementary altitude intervals (black curve), Butterworth low pass filter with a 8-km wavelength cutoff (Dotted black curve).

Figure 1 visualizes the spectral responses of the MRA using an 8th-order Daubechies wavelet with a six-level decomposition, the variance method, and the 5th-order Butterworth low-pass filter with an 8-km cutoff. The equivalent filter for the 5th detail of MRA decomposition is more selective than the Butterworth filter and comparable to the variance method filter (focus on 4 km vertical wavelength). Additionally, the Butterworth filter exhibits higher sensitivity to noise, particularly at wavelengths < 1 km. By adjusting the vertical sampling, MRA offers additional control over filter responses, allowing for tunable selectivity.

GW potential and kinetic energy densities (in J.kg$^{-1}$) can be computed from the formulas given in Wilson et al. (1991) :

$$E_p = \frac{1}{2} \left( \frac{g}{N} \right)^2 < \left( \frac{T'}{T_0} \right)^2 > \tag{5}$$

$$E_k = \frac{1}{2} < u'^2 + v'^2 > \tag{6}$$

Where $g$ is the Earth acceleration, $N$ the buoyancy frequency, $T_0$ the background temperature profile, $T'$, $u'$ and $v'$ are, respectively, temperature, zonal wind, and meridional wind perturbations ($w'$ is neglected).

The first step for computing GW energy densities is to determine the impact of the Gaussian white noise, present in the temperature and wind profiles, on the MRA decomposition. To do so, we simulated temperature profiles with gaussian white noise $\mathcal{N}(0, 1)$ and computed the potential energy of the gaussian white noise. A noise amplitude increasing with the altitude by a factor of $e^{\frac{z}{H}}$, where z is the altitude taken from 20 km to 80 km and H is the height scale (8 km), is here considered. Then, the mean energy is calculated on height intervals of 10 km for each detail from the MRA decomposition (level 6) of the 1000 gaussian white noise simulated signals to estimate the impact of the noise on the different spectral bands (Fig. 2).

The potential energy density of the noise is divided by 2 at each step of the decomposition which is a consequence of the pyramidal decomposition. Consequently, the noise-induced potential energy density calculated from $d_n$, $E_{p_n}$, can be estimated using an empirical formula, assuming that the first detail level of the MRA decomposition ($E_{p_1}$) consists solely of noise :
$E_{p_n} = \frac{E_{p_1}}{2^{n-1}}$, with n varies from 2 to 6 for a decomposition at order 6.



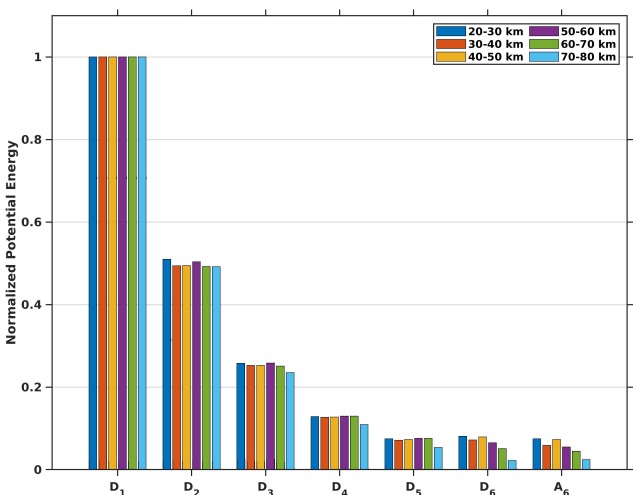

**Figure 2.** Normalized mean potential energy density of 1000 simulated Gaussian white noise signals for each detail level obtained from the MRA decomposition between 20 and 30 km km (blue), 30 and 40 km (orange), 40 and 50 km (yellow), 50 and 60 km (purple), 60 and 70 km (green), 70 and 80 km (cyan).

The GW energy density (GWED) is computed for each perturbation (temperature or wind) profile of the considered night of measurements. In order to subtract noise, we remove noise energy corresponding to the nth detail treated by applying the empirical formula. Then, the nightly mean profile of the GWE is determined, and the mean profile is smoothed using a 7.5 km Hanning window. This process enables an estimation of the primary propagation characteristics of GWs observed during the night. With the MRA decomposition, it is possible to derive an energy profile that focuses on a dominant or quasi-

monochromatic mode of gravity waves. It enables to highlight the interaction between modes of GWs during the night. Due to potential edge effects, the MRA method may bias GW energy densities as it approaches the upper or lower limits of vertical profiles, depending on the order of the detail and the amplitudes of the signal. To mitigate edge effects, it is recommended to either reduce the vertical range of the analyzed signal (e.g., analyzing the 30–70 km range instead of the full 20–80 km profile) or to apply signal symmetrization by extending the profile with a mirrored version of itself at both ends.

## 3  Comparison of the different methods

In the following section, we demonstrate the capability of the MRA method to estimate both background temperature and perturbation profiles by comparing its results with those obtained from conventional methods, including time-averaging, polynomial fitting, and spectral filtering. We further compare the GW potential energy derived from these four approaches with that obtained using the variance method described by Mzé et al. (2014), in order to evaluate their relative effectiveness. Lastly, we

apply the MRA method to wind profiles to investigate additional GW propagation characteristics. An illustrative case study is presented below.





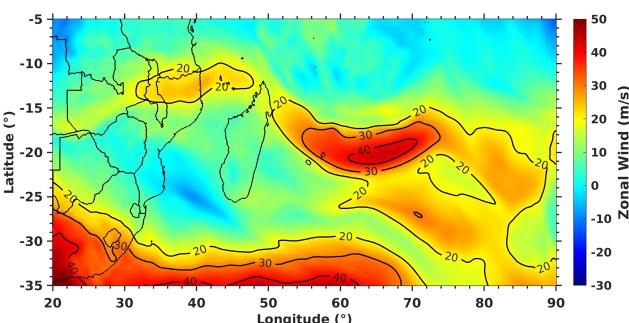

**Figure 3.** Horizontal Zonal Wind (m/s) at 200 hPa ( 13 km) on 20 November 2023 at 1800 UTC derived from ERA 5 reanalysis (Hersbach et al., 2020) above La Réunion (21.0°S,55,5°E). The subtropical jet is split into two main flows: one branch at latitudes of 30°S and the second one at latitudes between 10°S and 15°S. A jet core with wind intensity of 30-40 m/s is visualized above La Réunion (21°S, 55.5°E).

## 3.1 Case study: 20th November 2023

The GW energy densities derived from the MRA and conventional methods are illustrated using lidar profiles from the night of 20 November 2023. During this period, a split jet stream system was observed between 30°S and 10°S (Homeyer and Bowman, 2013), giving rise to instabilities and wind shear that contributed to pronounced GW activity in both the troposphere and stratosphere over La Réunion. This episode occurred during the late austral winter (20–24 November 2023), and ERA5 reanalysis data reveal a marked intensification of westerly winds over the island, associated with the passage of the upper-level jet stream (Fig. 3). In addition, prominent 24-hour tidal signatures were detected in the zonal and meridional wind components above 30 km, highlighting the presence of diurnal tidal waves in the upper stratosphere.

Vertical profiles of horizontal wind from ERA5 reanalysis (Hersbach et al., 2020) and radiosonde measurements on 20 November 2023 at 1200 UTC show good agreement and reveal the presence of short-scale GW structures (not shown). In particular, ERA5 wind profiles display clear stratospheric GW perturbations with an estimated vertical wavelength of approximately 5 km during the night of November 20, 2023. To further characterize these features, a hodograph analysis was performed by examining the altitude-dependent evolution of the horizontal wind components. It reveals an elliptical structure at heights of 30-35 km which characterizes a GW with a vertical wavelength of 5 km. The counter-clockwise rotation of the perturbation wind vector with height indicates upward energy propagation into the middle stratosphere. The axis ratio of the ellipse provides an intrinsic wave period of about 23 h. Assuming that the dominant GWs propagate primarily in the vertical direction and are detectable at higher altitudes in the lidar profiles, the following sections focus on a detailed characterization of these wave structures.

## 3.2 Application to lidar temperature profiles

First, the MRA method is evaluated against conventional approaches by comparing the estimation of background temperature, temperature perturbations, and GW potential energy density, using lidar temperature measurements acquired on 20 November



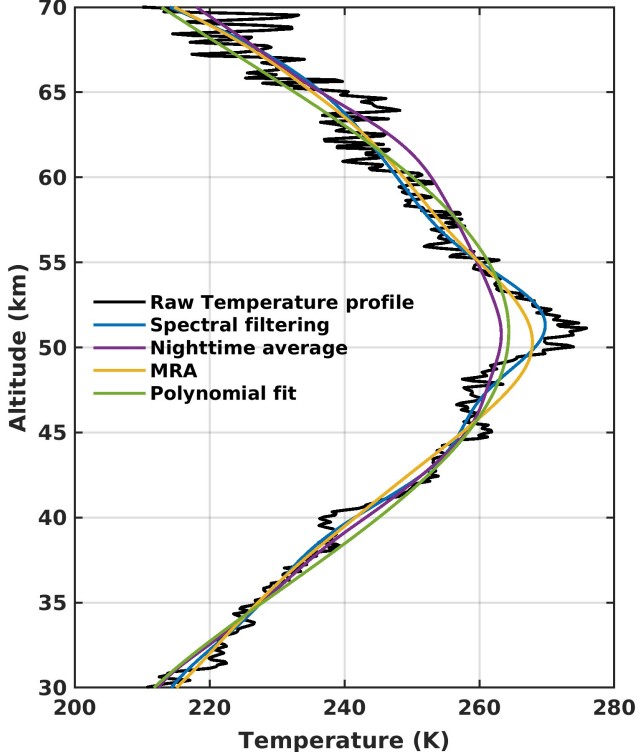

**Figure 4.** Original temperature profile at 1543 UTC (black), background temperature retrieved by different methods : Butterworth filter (blue), night mean profile (purple), trend ($a_6$) obtained from the MRA (yellow) and 4th order polynomial (green). The approximation $a_6$ corresponds to vertical wavelengths greater than 12.8 km which are not considered as GWs.

2023 between 1543 UTC and 2028 UTC, during a 5 hours observation period. Background temperature profiles were derived using spectral filtering with butterworth filter with cutoff wavelength at 8 km, polynomial fitting with 4th order polynomial function, nighttime averaging over the 5 hours of measurement, and the MRA method, applied to lidar temperature measurements from the night of 20 November (Fig. 4). Between 30 and 45 km altitude, all background estimates show similar values. However, near the stratopause, the backgrounds obtained using nighttime averaging and polynomial fitting are approximately 10 K lower than those produced by MRA and Butterworth filtering. Above 50 km, the background derived from nighttime averaging begins to show noise-induced features unrelated to GWs. In contrast, both the MRA and Butterworth filtering methods display consistent and smoother behavior, yielding a more realistic background structure at the stratopause level.

Based on the hodograph analysis, we focus on GWs with dominant vertical wavelengths around 5 km. To isolate these scales within the MRA framework, we use detail levels $d_4$ (1.6-3.2 km) and $d_5$ (3.2-6.4 km), which are suitable for capturing this wave structure. Perturbations extracted using different background removal methods reveal a distinct wave structure with a downward phase propagation (upward energy propagation) with a vertical wavelength of approximately 5 km at altitudes between 35 km and 45 km (Fig. 5 ). This feature appears more pronounced in the Butterworth-filtered temperature perturbations and the MRA-



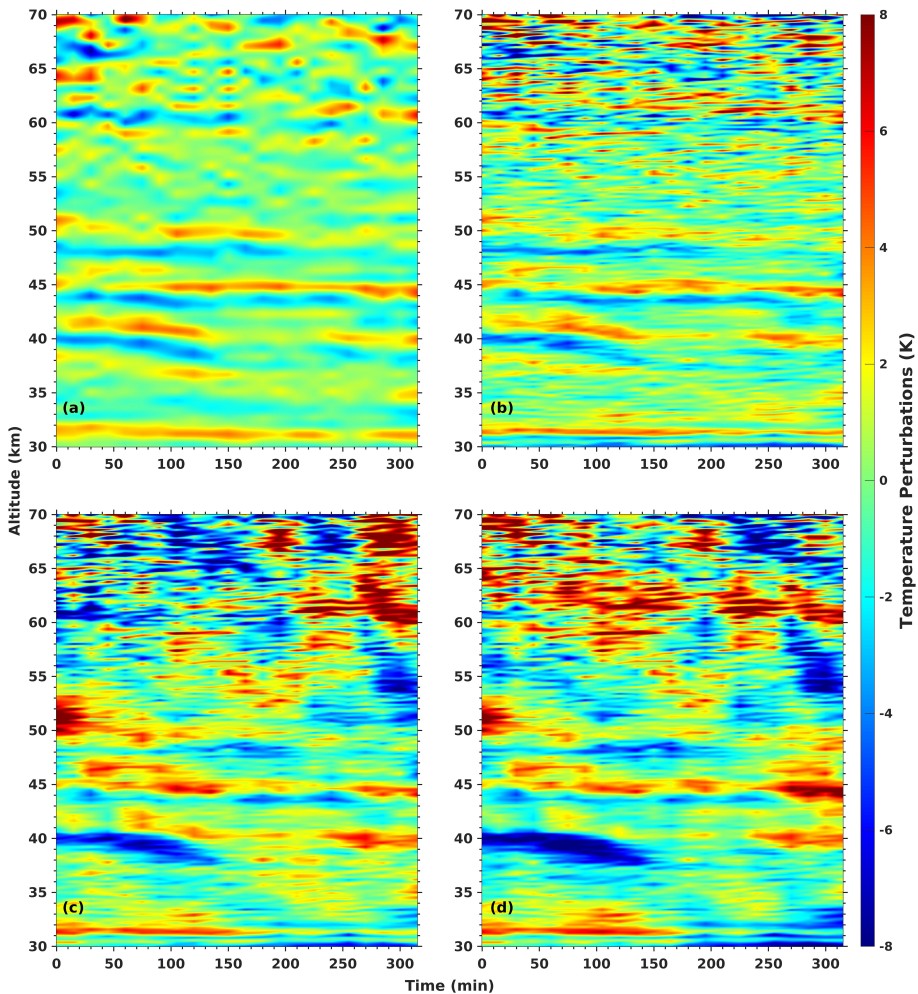

**Figure 5.** Height-time distribution of the temperature perturbations for the $20^{th}$ November 2023 night obtained from the four methods: MRA focused on $d_4$ (1.6 - 3.2 km) and $d_5$ (3.2 - 6.4 km) (a), Butterworth filter with a cutoff wavelength of 8 km (b), night mean (c), 4th order polynomial fit (d)

derived signals, highlighting the consistency of the MRA approach with conventional spectral filtering techniques. Above the stratopause (51 km), the Butterworth method emphasizes smaller-scale perturbations compared to MRA. In contrast, the night-mean and polynomial fit methods produce incoherent perturbation values at these heights (Figs. 5c, 5d). The MRA effectively reveals the 5-km vertical wavelength structure, which is clearly visible in the extracted perturbations. The results closely match those obtained using other background removal methods, particularly the Butterworth filter, which also highlights this wave component. However, the MRA is less sensitive to noise, resulting in improved signal clarity and a higher signal-to-noise ratio. These advantages underscore the effectiveness of the MRA in detecting GWs at this scale, making it a robust and reliable alternative to conventional filtering techniques.




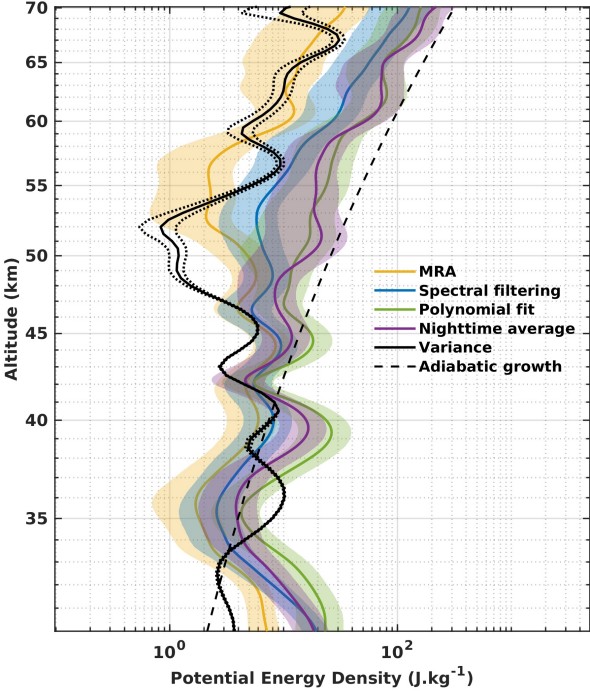

**Figure 6.** Mean GW potential energy density on the night of 20 November 2023, derived using different methods: variance method (black), MRA (yellow), Butterworth filter with an 8 km cutoff (blue), night-mean subtraction (purple), and polynomial fit (green). Dotted black lines represent the variance method error, dashed line represent the adiabatic growth, and shaded areas indicate the 95 % confidence intervals.

The main part of the study is the computation of the GW potential energy of which the variance method is taken as a
reference (Mzé et al., 2014). The GW potential energy density from the variance method computed for GW with 5 km vertical wavelength is taken as a reference. Vertical profiles of GW potential energy density derived from the MRA (focus on details $d_4$ and $d_5$), Butterworth filtering, night-mean, and polynomial fit methods exhibit similar trends up to 50-km height (Fig. 6).

Although the variance method profile follows the same overall trend, it exhibits an altitude shift. This offset arises because the variance method computes energy density directly from the raw lidar signal, whereas the other four methods rely on temperature
profiles that require additional assumptions during retrieval. Furthermore, since the raw signal is proportional to atmospheric density, a phase shift develops between temperature and density caused by the hydrostatic integration of the pressure profile from the bottom upward. This phase shift becomes more pronounced for GWs with longer vertical wavelengths. While the altitude shift can be determined for simulated monochromatic waves, it is more challenging to quantify when analyzing a full spectrum of GWs. In addition, the night-mean and polynomial fit profiles yield, on average, potential energy values roughly
twice as large as those derived from the MRA. Above 50 km altitude, the potential energy density profiles begin to diverge: whereas the MRA and Butterworth methods show a decrease in GW potential energy density, the night-mean and polynomial fit methods instead exhibit an increase. At 55-km height, the GWPE derived from MRA reaches its minimum, indicating GW





breaking or filtering at this altitude. The Butterworth method also detects this minimum, but at slightly lower heights and with GWPE values three to four times higher. In contrast, the night-mean and polynomial fit methods do not capture the minimum

and instead yield GWPE values ten times higher than those from MRA. Above 60-km height, the three conventional methods exhibit similar trends, with energy density values exceeding the expected density decrease, highlighting the impact of noise on the energy density profiles. Meanwhile, the MRA-derived GWPE profile closely follows the decreasing density curve.

Disregarding the altitude shift, the MRA method produces GW potential energy density values that are generally comparable to those obtained with the variance method between 30 km and 70 km altitude. The GWPE profile derived from the MRA also

mirrors the overall trends observed with the variance method: an increase between 30 km and 40 km, followed by a decrease to a minimum around 52.5 km, and then a renewed rise above 55 km, consistent with the expected exponential growth linked to the decrease in atmospheric density. Notably, near 55 km altitude, the MRA energy density profile shows values approximately twice as high as those of the variance method; however, the variance method's estimates remain within the MRA's 95 % confidence interval. Overall, despite the observed altitude shift, the MRA demonstrates strong agreement with the variance

method, highlighting its reliability in estimating GWPE.

### 3.3  Application to lidar wind profiles

Wind measurements obtained from lidar rely on calculating the phase shift of the Doppler signal. As a result, it is not possible to apply a variance-based method to derive kinetic energy density directly from wind lidar data. The earlier results have shown the effectiveness of the MRA approach for analyzing GWs. Applying the MRA to wind lidar profiles successfully isolates

perturbations with a vertical wavelength of around 5 km from detail coefficients $d_4$ and $d_5$. This wave pattern is clearly visible from the surface up to the highest observed altitudes in both zonal and meridional wind perturbations (Fig. 7 ).

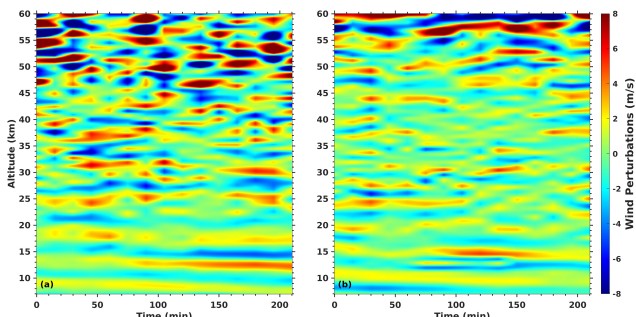

**Figure 7.** Height-time representation of zonal wind (a) and meridional wind (b) perturbations. Perturbations of wavelength ranging from 1.6 km to 6.4 km ($d_4$ and $d_5$) are represented in the height-time representation.

Figure 8a vizualises zonal and meridional wind perturbations lidar profiles from MRA detail $d_5$ ($\lambda_v$ = 3.2-6.4 km) on 20 November 2023. From 20 to 45 km, observed GW amplitudes are decreasing for both zonal and meridional winds and are increasing above 45 km. Phase quadrature is visible at altitudes between 51.1 km and 54.9 km, hence, a hodograph analysis

confirms the presence of an elliptical GW pattern with a vertical wavelength of  5 km at those altitudes (Fig. 8b). Moreover,




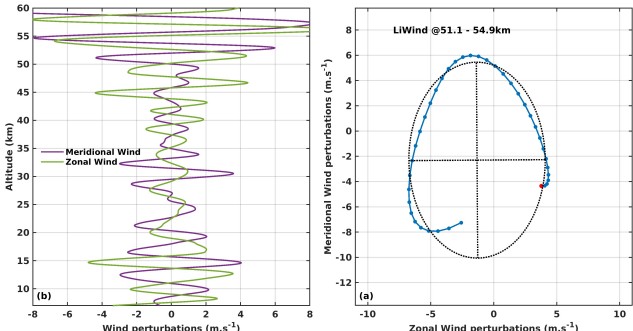

**Figure 8.** Zonal and meridional wind perturbations (a) and hodograph analysis (blue line) from 51.1 to 54.9 km (b) on 20 November 2023; Blue and red dots indicate height interval of 100 m and starting point at 51.1 km respectively. Dotted black lines represent the elliptical fit function and major and minor axes obtained by the least mean square method.

the orientation of the major axis suggests a horizontal propagation direction aligned north–south, perpendicular to the jet orientation. The anticlockwise rotation of the hodograph indicates upward propagation of GW energy into the mesosphere. The length and axis ratio of the ellipse further yield an estimated vertical wavelength of 5 km and an intrinsic period of around 23 hours.

To go further in understanding the GW propagation, the kinetic energy density from the MRA for each detail can be computed. As an example, the MRA method is applied on the wind lidar profile, and the mean kinetic energy density is computed on the night of 20 November 2023 (Fig. 9).

     At heights above 15 km, kinetic energy densities for details $d_2$, $d_3$, $d_4$ and $d_5$ increase up to 35 km, 50 km and 25 km respectively in contrast with the kinetic energy density for detail $d_6$ showing a decrease up to 30 km. However, from 25 km to 310   45 km, kinetic energy density for detail $d_5$ is decreasing. Nevertheless, kinetic energy densities for $d_2$ and $d_3$ ceases to increase at 35 km corresponding to higher values of kinetic energy for $d_6$. A connection between GWs with wavelengths between 0.4-1.6 km and 3.2-6.4 km and larger GWs with wavelengths between 6.4-12.8 km is highlighted by the result of the MRA. Above 45 km, kinetic energy for $d_4$, $d_5$ and $d_6$ is increasing whereas the kinetic energy for $d_2$ and $d_3$ still decreasing.

     The ratio of kinetic energy density to potential energy density was computed for three altitude intervals 30-40 km, 40-50 315   km, and 50-60 km for each detail of the MRA decomposition (Fig. 10). For details $d_2$, $d_3$, $d_5$, and $d_6$, this ratio increases from 30-40 km to 40–50 km, whereas for detail $d_4$ it decreases over the same altitude range. From 40-50 km to 50-60 km, the ratio decreases for the smaller vertical wavelengths ($d_2$ and $d_3$) but increases for the larger vertical wavelengths ($d_4$, $d_5$, and $d_6$). This evolution reflects the modification of the GWs frequency distribution with altitude (Wilson et al., 1990). Between the stratosphere and lower mesosphere (approximately 40-60 km), low-frequency GWs are increasingly damped, leading to a 320   decreasing energy ratio for these components at higher altitudes. In contrast, high-frequency GWs continue to propagate into the mesosphere, maintaining or increasing their energy ratio. Furthermore, the increase in the ratio for $d_4$, $d_5$, and $d_6$ with altitude can be interpreted as a shift in the nature of GWs. As low-frequency waves are filtered out in the lower mesosphere,





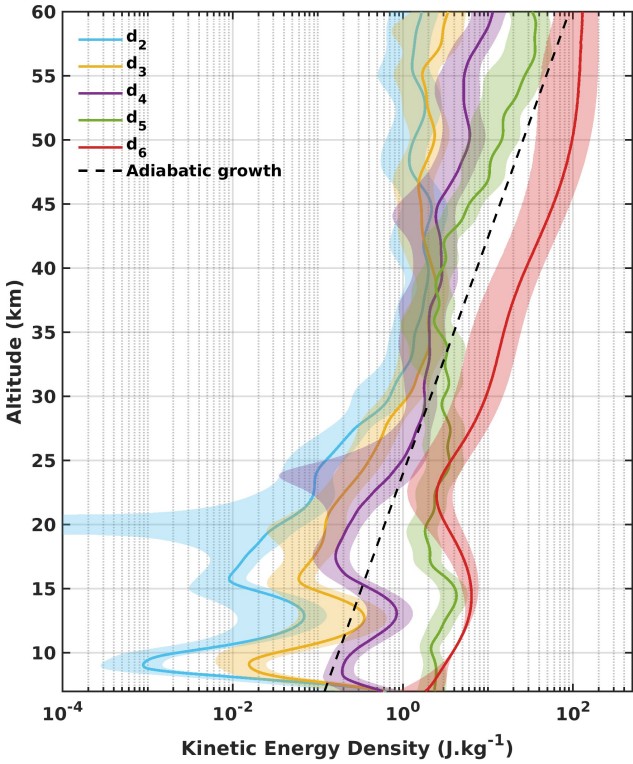

**Figure 9.** Mean GW kinetic energy density (b) during the night of 20 November 2023. Energy densities are computed from perturbations extracted using MRA decomposition at different scales: $d_2$ (0.4-0.8 km, blue ), $d_3$ (0.8–1.6 km, yellow), $d_4$ (1.6–3.2 km, purple), $d_5$ (3.2–6.4 km, green), and $d_6$ (6.4–12.8 km, red). Shaded areas indicate the 95 % confidence intervals.

the intrinsic frequency of the remaining GWs approaches the Coriolis frequency, suggesting that inertia-gravity waves become dominant in the mesosphere.

## 4   Summary, conclusions and perspectives

This study presented the implementation of a method based on the MRA perfomed with the $8^{t}h$ order Daubechies wavelet to investigate GWs. The approach was evaluated by comparing results obtained from MRA with those from conventional methods, including night-mean subtraction, polynomial fitting, and spectral filtering. The analysis is illustrated on a case study from 20 November 2023, when a split jet structure, along with associated instabilities and wind shear, generated significant GW activity in the troposphere above La Réunion. The decomposition provided by MRA enabled spatial and spectral localization of GW structures along individual lidar profiles. When applied to temperature lidar measurements, MRA showed strong agreement with the Butterworth spectral filtering in estimating the background field. While the dominant structure of 5-km vertical wavelength was also visible in perturbations extracted by conventional methods, those methods proved more susceptible to




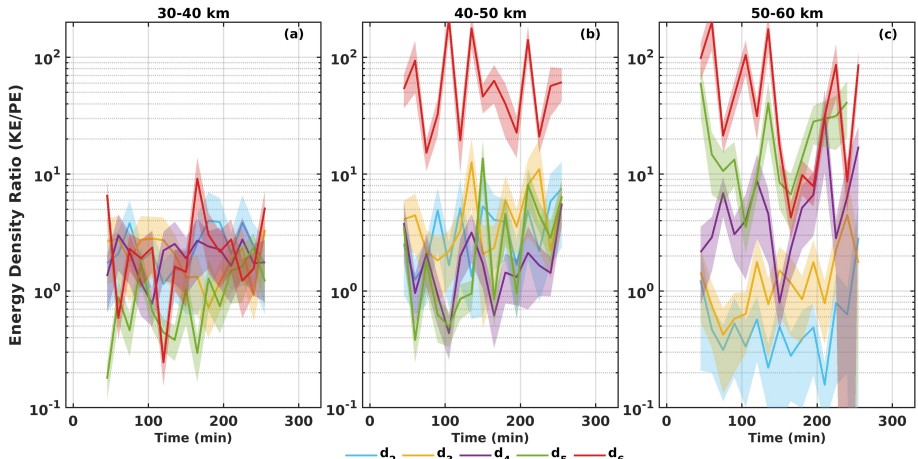

**Figure 10.** Energy density ratio (KE/PE) computed for MRA details $d_2$ (blue), $d_3$ (yellow), $d_4$ (purple), $d_5$ (green) and $d_6$ (red) during the 20 November 2023 night between 30-40 km (a); 40-50 km (b) and 50-60 km (c). Shades represent the uncertainty error.

noise. By contrast, MRA offered more precise space(in terms of altitude) and time (during the observation period) localization

and highlighted multiscale GW structures. Additionally, comparison of GW potential energy profiles derived from MRA and from the variance method underscored an altitude shift introduced by integrating density to retrieve temperature profiles which is dependent on the GWs vertical wavelength. To deepen the understanding of GW propagation from their generation in the lower atmosphere to their breaking in the middle atmosphere, MRA enables the calculation of energy density profiles (both kinetic and potential energy densities) for each detail, corresponding to tunable vertical wavelength bands with octave or dyadic

wavelet decomposition. These energy density profiles reveal how GW energy density is distributed across scales, and computing the kinetic-to-potential energy density ratio across different altitude ranges offers insights into the vertical evolution of GW frequency distribution and intrinsic wave characteristics. The combined multi-scale energy density profiles from MRA also reveal interactions between different GW scales, which are harder to detect using other approaches. Looking ahead, statistical analyses and climatological studies of larger lidar datasets could be used to characterize GW propagation more systematically

above La Réunion. Extending the study to include the troposphere, through the integration of radiosonde data or reanalysis outputs, could help identify GW properties linked to their specific sources such as tropical cyclones during austral summer. Since GW characteristics depend on their generation mechanisms and the way they interact as they propagate, this multi-scale perspective is especially valuable. However, it is important to acknowledge the inherent limitations of lidar observations, which are primarily one-dimensional and mostly available during nighttime, limiting the reconstruction of horizontal wave structures

and full 3D GW dynamics. Future observational strategies should aim to combine lidar measurements with complementary instruments such as radiosondes, satellite sounders, or GNSS radio occultation profiles. Such multi-instrument synergy could provide a more complete and nuanced view of GW propagation and vertical energy transfer in the atmosphere.



*Code availability.* The multiresolution analysis implemented in this study was developed in MATLAB, relying on functions from the Wavelet Toolbox.

*Data availability.* The OPAR lidar data can be obtained through the NDACC database at https://ndacc.larc.nasa.gov/. The post-processed data used in this study are available upon request from the principal investigators: AH for the temperature lidar and SK for the wind lidar. ERA5 model-level reanalysis data were obtained from the Copernicus Climate Change Service (C3S) Climate Data Store using the CDS API. The dataset used is: "ERA5: Fifth generation of ECMWF atmospheric reanalyses of the global climate" (DOI: https://doi.org/10.24381/cds.143582cf). The data were generated by the European Centre for Medium-Range Weather Forecasts (ECMWF).

*Author contributions.* ST, FCM conceived the study. AH and SK post-processed the temperature and wind lidar data respectively. FCM, AH and PK offered scientific insights. The paper was written by ST with contributions of all authors.

*Competing interests.* The authors declare that they have no conflict of interest.

*Acknowledgements.* The authors thank scientific and technical teams who contributed to the data acquisition at the Maïdo Observatory (https://www.osureunion.fr/). We acknowledge the use of ERA5 reanalysis model-level data provided by the Copernicus Climate Change Service (C3S), generated by ECMWF.



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
