# Peer review of "Implementation of a Multi-resolution Analysis Method to Characterize Multi-Scale Wave Structures in Lidar Data"

_EGUsphere, 2025_

## Referee Comment (RC1)

Review of **"Implementation of a Multi-resolution Analysis Method to Characterize Multi-Scale Wave Structures in Lidar Data"** by Samuel Trémoulu et al. (2025)

**General comments**

The authors apply multi-resolution analysis (MRA) using the 8th-order Daubechies wavelet to lidar measurements of the middle atmosphere at La Réunion. While MRA was originally developed more than three decades ago, its application to lidar data in the present form is novel. The authors compare MRA-filtering to several commonly used filtering techniques:

- (A) nightly mean subtraction
- (B) polynomial fitting in the vertical
- (C) spectral filtering in the vertical
- (D) a variance method

This comparison is, in principle, of interest. However, the study lacks a clearly defined benchmark or metric that allows a systematic evaluation of the different approaches. The manuscript states that it "presents a method based on MRA to characterize multi-scale GWs in observational data" and that "in section 3, the characteristics as well as the comparative performance of the four methods are discussed." Yet, the overall aim remains ambiguous. What exactly is being characterized? Is it GW amplitudes, wavelengths, periods, localization, GWPE/GWKE? How are these aspects systematically compared across methods?

In section 3.2, the authors present background temperature profiles, GWPE profiles, and time–height sections of perturbations filtered by different techniques. Section 3.3, however, focuses solely on MRA-filtering. Why is there no systematic comparison here? In particular, for the hodograph analysis, it would be valuable to assess how the choice of filtering method influences the outcome and physical interpretation.

For clarity, I prepared a table summarizing the spectral properties of the different methods (E = MRA, details 3–6):

|            | A)        | B)    | C)   | D)       | E)          |
|------------|-----------|-------|------|----------|-------------|
| period     | <4h 45min? | all   | all  | ???      | All         |
| wavelength | <7.5km    | <26km | <9km | 2.5-6km? | 0.8 - 12.8km |

Figure 1 illustrates the spectral response of several filters, but responses for nightly mean and polynomial subtraction are missing. Please include them for completeness. A normalized FFT spectrum would also help to indicate where wave energy is expected. Either show spectral responses of *all* filters in Figure 1, or summarize their passbands in a table for reader convenience.
* * *
**Specific comments**

**Section 1 Introduction**

- Very well written and nicely funnels down from "GWs are important" to "we need Multi-resolution analysis to study GW observations".
- Please sharpen the aim of the study. Construct a proper working hypothesis or state a research question.

**Section 2.2 (GW analysis techniques)**

- Derive and define GWPE and GWKE (without the "density" term) here, stressing the importance of proper background–perturbation separation.
- Clarify the meaning of the brackets in Eqs. (5) and (6); these should represent averages over at least one vertical wavelength.

**2.2.1 Nightly mean subtraction (A)**

- Subtracting the nightly mean can strongly reduce stationary wave structures (over the ~4h45min observation window). Presumably, this motivated the Hann smoothing step.

Please justify the chosen window size (7.5 km). How general is this choice? Would you increase it (e.g., to 15 km) for quasi-stationary mountain waves?

**2.2.2 Polynomial fit subtraction (B)**
- The statement that an nth-order polynomial removes perturbations longer than a fraction of the height range is problematic: polynomials are spectrally broad, not trigonometric. Please provide a reference.
- Why was a 4th-order polynomial chosen (implying a cutoff at >26 km)? Was the fit weighted by temperature uncertainties? Why not use higher orders to align the cutoff with other filters?

**2.2.3 Spectral filtering (C)**
- The vertical cutoff is clear, but what is the impact in the temporal domain? Was the cutoff chosen to match MRA detail 5?

**2.2.4 Variance method (D)**
- The parameter $N_z = 21$ is only revealed in the caption of Fig. 1. Combined with the vertical resolution, this suggests sensitivity to structures >3.15 km. What limits amplitudes beyond ~6 km? What about temporal averaging ($N_t$)?

**2.2.5 MRA filtering (E)**
- Details 3–6 capture wavelengths 0.8–12.8 km. Again, what are the implications in the temporal domain?
- Why assume that noise is contained solely in detail 1? White noise is expected across all scales.

**Section 3.1 Case study**
- This section mainly confirms, via ERA5, the presence of GWs. Mentioning tides is misleading and should be removed.
- Instead of a 200 hPa map of zonal wind, show ERA5 time–height cross sections of temperature/wind perturbations at La Réunion. The current map does not allow the reader to locate the island.

**Section 3.2 Temperature profiles**
- Include an ERA5 temperature profile in Fig. 4 for reference.
- Add subpanels in Fig. 5 showing the subtracted background (and possibly noise). Use a diverging colormap for perturbations.
- The oscillatory behavior in GWPE (30–45 km, Fig. 6) likely stems from Hann smoothing. A boxcar filter may mitigate this. Also correct the unequal scaling of altitude in the y-axis.
- Conservative growth is only discussed above 50 km, but deviations below that altitude are also worth analyzing.
- The claim that MRA is less sensitive to noise requires explanation. Why is this the case relative to other methods? For fairness, low-order details (1–2) should be included in the comparison.

**Section 3.3 Wind profiles**
- The sudden introduction of hodograph analysis is not well motivated. If retained, it should be applied systematically across all methods to reveal differences.
- Variability of GWKE below 15 km is not discussed. Larger scales contain more energy, but one might expect maximum energy for detail 5.
- The GWPE/GWKE ratio is computed only for MRA. This should be repeated for all methods to evaluate potential misinterpretations caused by their differing spectral passbands.

**Section 4 Summary and perspectives**
- Provide stronger evidence of wave activity (e.g., ERA5 profiles at the lidar site).
- Clarify that discrepancies between methods primarily arise from their different spectral passbands.
- Emphasize that while the choice of wavelet and sampling defines the MRA bands, this is conceptually no different from setting bandpass filters.

- Highlight more clearly the advantage of MRA: the orthonormal decomposition allows energy-conserving reconstruction of GW-induced perturbations.

**Recommendation**

The manuscript introduces a potentially valuable approach. However, substantial clarifications and additional analyses are required before the work can be considered for publication.

---

## Author Comment (AC1)

**Response to Referee 1**

All authors sincerely thank Referee 1 for the time and care spent reviewing this manuscript, and for the constructive comments provided. Please find our detailed, point-by-point responses.

**General Comments:**

The authors apply multi-resolution analysis (MRA) using the 8th-order Daubechies wavelet to lidar measurements of the middle atmosphere at La Réunion. While MRA was originally developed more than three decades ago, its application to lidar data in the present form is novel.

The authors compare MRA-filtering to several commonly used filtering techniques:

- (A) nightly mean subtraction
- (B) polynomial fitting in the vertical
- (C) spectral filtering in the vertical
- (D) a variance method

This comparison is, in principle, of interest. However, the study lacks a clearly defined benchmark or metric that allows a systematic evaluation of the different approaches. The manuscript states that it "presents a method based on MRA to characterize multi-scale GWs in observational data" and that "in section 3, the characteristics as well as the comparative performance of the four methods are discussed." Yet, the overall aim remains ambiguous. What exactly is being characterized? Is it GW amplitudes, wavelengths, periods, localization, GWPE/GWKE? How are these aspects systematically compared across methods? In section 3.2, the authors present background temperature profiles, GWPE profiles, and time height sections of perturbations filtered by different techniques. Section 3.3, however, focuses solely on MRA-filtering. Why is there no systematic comparison here? In particular, for the hodograph analysis, it would be valuable to assess how the choice of filtering method influences the outcome and physical interpretation.

We thank the reviewer for this valuable comment. The objectives of the study have been clarified in the introduction, specifying that the goal is to evaluate the ability of MRA to isolate and characterize GW perturbations and associated energy (GWPE and GWKE) relative to conventional methods. A systematic quantitative comparison based on the relative difference of GW energies has been added. Section 3.3 was revised to include results from the conventional approaches for wind perturbations, GWKE, and the kinetic-to-potential energy ratio, while the hodograph analysis was removed to maintain a consistent comparison framework.

For clarity, I prepared a table summarizing the spectral properties of the different methods (E = MRA, details 3–6):

|            | A)     | B)    | C)   | D)       | E)          |
|------------|--------|-------|------|----------|-------------|
| Period     | > 1h   | > 1h  | > 1h | > 1h     | > 1h        |
| Wavelength | <7.5km | <10km | <9km | 2.5-6 km | 0.8-12.8 km |

Figure 1 illustrates the spectral response of several filters, but responses for nightly mean and polynomial subtraction are missing. Please include them for completeness. A normalized FFT spectrum would also help to indicate where wave energy is expected. Either show spectral responses of all filters in Figure 1, or summarize their passbands in a table for reader convenience.

As suggested, Figure 1 has been replaced by Table 1, which summarizes the spectral properties of the different methods.

**Specific comments**

**Section 1 Introduction**

- Very well written and nicely funnels down from "GWs are important" to "we need Multi resolution analysis to study GW observations".
- Please sharpen the aim of the study. Construct a proper working hypothesis or state a research question.

As requested, we have revised the introduction (lines 94-115) to clarify the aim of the study and explicitly state our research questions and working hypothesis.

**Section 2.2 (GW analysis techniques)**

- Derive and define GWPE and GWKE (without the "density" term) here, stressing the importance of proper background–perturbation separation.
- Clarify the meaning of the brackets in Eqs. (5) and (6); these should represent averages over at least one vertical wavelength.

We agree that the bracket notation was potentially ambiguous. We changed the brackets to an overline that represents the nightly average (line 220-222).

**2.2.1 Nightly mean subtraction (A)**

• Subtracting the nightly mean can strongly reduce stationary wave structures (over the ~4h45min observation window). Presumably, this motivated the Hann smoothing step. Please justify the chosen window size (7.5 km). How general is this choice? Would you increase it (e.g., to 15 km) for quasi-stationary mountain waves?

The 7.5 km Hanning window is chosen to optimize the signal-to-noise ratio for long-term climatological studies. For our purpose of extracting GWs, this effectively acts as a low-pass filter, defining the background state as structures with vertical scales larger than ~7.5 km.

We acknowledge this choice influences the extracted wave spectrum and have clarified this in the methods section. For specific cases like quasi-stationary mountain waves, a larger window might be necessary, but for the propagating non-orographic waves in this case study, this choice is appropriate for separating the background.

**2.2.2 Polynomial fit subtraction (B)**

consistency between the described approach and the text.

• The statement that an nth-order polynomial removes perturbations longer than a fraction of the height range is problematic: polynomials are spectrally broad, not trigonometric. Please provide a reference. We adapt the method by applying a sliding polynomial fit with the Savitzky–Golay filter to ensure

The text has been modified at line 161:" The Savitzky-Golay filter method,..."

• Why was a 4th-order polynomial chosen (implying a cutoff at >26 km)? Was the fit weighted by temperature uncertainties? Why not use higher orders to align the cutoff with other filters?

We applied a sliding polynomial fit using a Savitzky–Golay filter, selecting a third-order polynomial with a window width of 10 km, corresponding to an effective cutoff wavelength of approximately 5.5 km. The Savitzky–Golay filter is mathematically equivalent to a convolution filter whose coefficients are obtained from an unweighted least-squares regression. As a result, the fit does not incorporate temperature measurement uncertainties, and a constant noise variance is assumed along the profile. Guest et al. (2000) reported that polynomial orders higher than 3–4 can remove or distort inertia–gravity wave signals. Accordingly, our analysis was restricted to a third-order fit. Although increasing the polynomial order could, in principle, shift the effective cutoff toward shorter vertical wavelengths, such adjustments would risk filtering out parts of the gravity-wave spectrum relevant to this study and were therefore considered inappropriate.

**2.2.3 Spectral filtering (C)**

• The vertical cutoff is clear, but what is the impact in the temporal domain? Was the cutoff chosen to match MRA detail 5?

Spectral filtering was applied only in the spatial domain; therefore, it has no global impact on the temporal domain. The cutoff was selected to match the MRA details 4 and 5.

**2.2.4 Variance method (D)**

• The parameter Nz = 21 is only revealed in the caption of Fig. 1. Combined with the vertical resolution, this suggests sensitivity to structures >3.15 km. What limits amplitudes beyond ~6 km? What about

**temporal averaging (Nt)?**

The effective vertical passband of the variance method is determined by the parameter Nz and the corresponding vertical extent DZ = Nz\*dz\$. For Nz = 21 and dz = 150 m, the resulting filter behaves as a band-pass with a full width at half maximum (FWHM) sensitivity between approximately 2.5 km and 6 km, consistent with the spectral response shown in Figure B1 of  $Mz\acute{e}$  et al. (2014). In this configuration, the amplitude response decreases for wavelengths longer than about 6 km. However, the variance method exhibits a slower decay of its spectral response with increasing wavelength (and with increasing Nz), leading to weaker attenuation of large-scale perturbations and thus lower spectral selectivity compared to the MRA. Temporal averaging over 15-minute intervals (Nt = 15) is applied, and the resulting energy profiles are then averaged over the entire night of measurements.

**2.2.5 MRA filtering (E)**

• Details 3–6 capture wavelengths 0.8–12.8 km. Again, what are the implications in the temporal domain?

The temporal domain is unaffected, as all filtering operations are confined to the spatial domain.

• Why assume that noise is contained solely in detail 1? White noise is expected across all scales.

First, the data were oversampled to a vertical resolution of 100 m. The original temperature profiles from lidar measurements have a vertical resolution of 150 m, allowing the detection of gravity waves with minimum vertical wavelengths of approximately 300 m. After oversampling, the first detail level of the MRA corresponds to vertical wavelengths between roughly 200 m and 400 m. This interval contains only white noise, as gravity waves with vertical wavelengths shorter than 300 m cannot be resolved by the instrument.

While white noise is present at all scales, the dyadic structure of the MRA progressively reduces its contribution to the total energy (noise+GWs) at higher decomposition levels: the noise energy is divided by a factor of  $2^{n-1}$  at the  $n^{th}$  level. This behavior is illustrated in Figure 2, which shows the normalized white noise energy decreasing by a factor of two at each decomposition step, assuming that  $d_1$  contains only noise. This property of MRA provides a straightforward way to estimate and subtract the noise contribution across scales.

**Section 3.1 Case study**

• This section mainly confirms, via ERA5, the presence of GWs. Mentioning tides is misleading and should be removed.

The mention of tides is removed.

• Instead of a 200 hPa map of zonal wind, show ERA5 time—height cross sections of temperature/wind perturbations at La Réunion. The current map does not allow the reader to locate the island.

The previous figure has been replaced by a new one displaying temperature, zonal wind, and meridional wind perturbations over La Réunion (Figure 2), which clearly reveal the presence of gravity waves, especially on 20 November 2023 in the stratosphere.

**Section 3.2 Temperature profiles**

• Include an ERA5 temperature profile in Fig. 4 for reference.

Figure 4 has been renumbered to figure 3 and we add the ERA5 temperature profile selected at the closest UT time to the lidar profile.

• Add subpanels in Fig. 5 showing the subtracted background (and possibly noise). Use a diverging colormap for perturbations.

The figure has been updated as Figure 4, using only undenoised signals to estimate perturbations from the spectral filtering, sliding polynomial fit, and night-average methods. Consequently, the subpanel representing the background without noise correction corresponds to the night-average method. A diverging colormap was applied to display the perturbations.

• The oscillatory behavior in GWPE (30–45 km, Fig. 6) likely stems from Hann smoothing. A boxcar filter may mitigate this. Also correct the unequal scaling of altitude in the y-axis.

The oscillations in GWPE (Fig. 5) do not result from the Hann smoothing. The Hann window was applied solely to ensure consistency with the smoothing used in the variance method. In addition, the unequal altitude scaling on the y-axis has been corrected.

• Conservative growth is only discussed above 50 km, but deviations below that altitude are also worth analyzing.

Deviations between the methods in GWPE are discussed below 50 km. In addition, a systematic comparison was introduced using the relative difference with respect to the MRA.

• The claim that MRA is less sensitive to noise requires explanation. Why is this the case relative to other methods? For fairness, low-order details (1–2) should be included in the comparison.

This point is relative to the previous points on Fig. 5. We didn't mention that we use denoised signals (by removing detail 1 of the MRA) to compute gravity waves perturbations for the spectral filtering, polynomial fit and night average method. We changed Fig. 5 and adapted the text.

**Section 3.3 Wind profiles**

• The sudden introduction of hodograph analysis is not well motivated. If retained, it should be applied systematically across all methods to reveal differences.

The hodograph of wind perturbations has been removed.

• Variability of GWKE below 15 km is not discussed. Larger scales contain more energy, but one might expect maximum energy for detail 5.

Variability of GWKE below 15 km is discussed at line 352. We changed the background estimation for the MRA so the energy of detail 6 will not appear.

• The GWPE/GWKE ratio is computed only for MRA. This should be repeated for all methods to evaluate potential misinterpretations caused by their differing spectral passbands.

The GWPE/GWKE ratio is computed for MRA, spectral filtering, nighttime average (As mentioned in the article, the variance method is not available for wind lidar measurements). We kept the ratio computed for details  $d_2$ ,  $d_3$ ,  $d_4$  and  $d_5$ .

**Section 4 Summary and perspectives**

- Provide stronger evidence of wave activity (e.g., ERA5 profiles at the lidar site).
- Clarify that discrepancies between methods primarily arise from their different spectral passbands.
- Emphasize that while the choice of wavelet and sampling defines the MRA bands, this is conceptually no different from setting bandpass filters.

Indeed, MRA is conceptually similar to applying a set of bandpass filters. However, the choice of wavelet and the sampling scheme strongly influence the behavior of the MRA bands. Changing the wavelet type can bias the amplitude of the retrieved GWs (Guo et al., IEEE, 2022). Daubechies wavelets are commonly used for signal reconstruction, and their frequency-domain localization improves with increasing order. Chane Ming et al. (2000) demonstrated that the Daubechies wavelet of order 8 is well suited for the decomposition and reconstruction of GW perturbations, ensuring a reliable estimation of GW activity.

• Highlight more clearly the advantage of MRA: the orthonormal decomposition allows energy-conserving reconstruction of GW-induced perturbations.

The *Summary and Perspectives* section was revised to clarify the comparison of methods, their limitations, and the benefits of the MRA approach, as suggested by the reviewer.

**Recommendation**

The manuscript introduces a potentially valuable approach. However, substantial clarifications and additional analyses are required before the work can be considered for publication.

---

## Author Comment (AC2)

**RESPONSE TO REFEREE 2**

All authors sincerely thank Referee 2 for the time and care spent reviewing this manuscript, and for the constructive comments provided. Please find our detailed, point-by-point responses.

This study presents a multi-resolution analysis (MRA) approach using the 8th-order Daubechies wavelet to analyse gravity waves in lidar data for a single case event. The MRA is compared with traditional methods such as mean background atmospheric temperature, polynomial fitting, Butterworth spectral filtering, and the variance method. The authors demonstrate that MRA outperforms these methods when applied to lidar temperature data. For wind analysis, only MRA is used.

This work introduces a promising tool for gravity wave analysis that could contribute to the understanding of multi-scale atmospheric dynamics. The manuscript is well written; however, it requires additional clarification and explanation of certain choices.

**General comments:**

- Why are the chosen windows different between methods? It needs some justification. We aimed to match the passbands of the different methods as closely as possible to isolate the dominant GWs with a vertical wavelength of 5 km. This point is emphasized in lines 275-277.
  - It would be useful to see a comparison of the MRA to a conventional wind analysis.

We updated the results presented in the section "Application to wind lidar profiles" by including additional outputs from the conventional analysis methods. Specifically, we added results on wind perturbations, gravity wave kinetic energy (GWKE), and the kinetic-to-potential energy ratio to provide a more comprehensive comparison with the MRA results.

• Section 3.1 Case Study: Several mentions are made of wind, tides, and GWs observed on the day, but the corresponding figures are not shown. Since the paper focuses on the method and the case study, including these figures would help the reader follow the discussion without having to rely solely on the authors' statements.

As also noted by Reviewer 1, we have added Figure 2, which shows gravity wave activity over La Réunion from 20 to 25 November 2023, derived from ERA5 analyses.

• Throughout the article, the same method is called differently, e.g. 'butterworth filter' and 'spectral filtering'. Select one way of calling it and use it everywhere.

We use 'spectral filtering' throughout the entire article.

There is a misuse of GW and GWPE throughout the entire article. GWPE is defined but then
used as GW potential energy. Please change it everywhere, and remove densities after
GWPE.

Modifications have been done. We defined and used GWPE and GWKE.

• The summary, conclusions and perspectives need some emphasis on the method comparisons and their weaknesses.

The summary, conclusions and perspectives section has been revised with clearer discussion of the method comparisons, their limitations, and the advantages of MRA.

**Specific comments:**

Figure 4: It could be useful to see a plot with the difference between the temperature and method next to the temperature one.

Figure 4 has been renumbered as Figure 3, and we have added a new subplot showing temperature perturbations derived by removing the background from the original profile for each method.

L007: In an abstract, you should try to avoid the use of acronyms to help readability. It's okay to use them; however, don't use an acronym for defining another acronym GW potential energy (GWPE). You are not space-limited, so avoid this practice.

L010 : The sentence has been changed into "In terms of gravity waves potential energy (GWPE)..."

L017-021: Paragraph small, only one citation.

The paragraph has been changed (L021 - L031) and we add new citations.

L023: Here the 'e.g.' is used. If you are giving an example, only one or two citations are enough. It could be one old and a new one. Instead, 10 are given and not even one is a recent one (2020+). I suggest removing e.g. and adding a later one too.

The 'e.g.' has been removed. We add a later citation (L035).

L024: Duck et al. 2001, the link is broken.

The DOI link is now active.

L025: I agree that Lidar observations are capable of inferring long-term trends; however, some of the cited publications are campaigns or techniques, which aren't studies in long-term trends.

Additional references to studies on long-term trends have been included.

L028-29: Same as before, there are newer studies of GW with Lidar observations.

We have added citations of newer studies of GW with lidar observations (L039-40).

L085: cal/val is only used twice and in the same sentence. If it is not used later, it is unnecessary to define it.

L107: The sentence was revised to: "Lidar data are primarily used for long-term monitoring of the middle atmosphere and have also supported recent calibration activities, such as those conducted for the European Space Agency's ADM-Aeolus satellite mission dedicated to global wind observations"

L150: verticalsinterval

L171: This error has been corrected in the revised manuscript.

L163/164: You defined MRA in the second sentence but use it in the first sentence, please switch.

L184/185: The modification has been done.

L190: eq:3 is the equation of the gravity wave potential energy. Remove density and all the references to it. You have already defined GWPE, so now you can use it.

We have used GWPE.

L200: The reference to the figure is there, but no in-text full explanation of the figure.

For improved clarity, we have replaced the figure with a table that summarizes the vertical wavelength intervals corresponding to all methods.

Fig2: caption double km.

Figure 2 has been renumbered to Figure 1 and the caption has been modified.

L205: Again, defining an acronym with an acronym and it should be defined before. Particularly in this case, you don't need it because it's the GWPE.

We define GWKE as Gravity Wave Kinetic Energy.

L207: GWE?

We have replaced the abbreviation "GWE" with its full form, "Gravity Wave Energy," for improved clarity.

Fig. 3: second line: "above" La Réunion, it sounds narrower than actually is. Change to centred over or similar. And the longitude is written with a ","

Figure 3 has been renumbered as Figure 2. We have replaced Figure 2 with a height-time cross section of temperature and wind perturbations from ERA5 over La Réunion, which confirms the presence of gravity wave structures above the island.

Figure 8: red dots? Only one is shown at which I guess is the start.

In accordance with Reviewer 1's request, we have removed the hodograph analysis.

Figures 7 and 8: Increase label size.

Figure 7 has been renumbered as Figure 6, and its labels have been resized. Additionally, Figure 8, presenting the hodograph analysis, has been removed from the article.